# Using a Metadata Approach to Extend the Functional Resonance Analysis Method to Model Quantitatively, Emergent Behaviours in Complex Systems

Rees Hill [1] and David Slater [2,*]

1 Zerprize Ltd., Christchurch 8053, New Zealand; rees.hill@zerprize.co.nz
2 School of Engineering, Cardiff University, Cardiff CF24 3AA, UK
* Correspondence: dslater@cambrensis.org

**Abstract:** In an increasingly complex world there is a real, urgent need for methodologies to enable engineers to model complex sociotechnical systems, as these now seem to describe the majority of systems in use today. This is, of course, exacerbated by the increasing involvement and augmentation with "black box" AI contributions. Hollnagel produced a methodology (FRAM) which did allow the analyst insights into these systems' behaviour, but the model-based system engineering applications demand numbers and a quantitative approach. In the last 10 years, this original approach, developed to model systems as sets of interactive, interdependent "functions" (abstracted from agent or component details), has been further developed to the point where it can take the basic data and structures from the current component-focussed system engineering "models", and can pull them all together into dynamic models (as opposed to the static, fixed System Theoretic Process Accimaps) from which analysts can discern how they really work in practice, and predict the emergent behaviours characteristic of complex systems. This paper describes how the FRAM methodology has now been extended to provide these extra, essential attributes. It also describes its implementation using an open-source software, freely available for use and verification on the GitHub site.

**Keywords:** complex systems; model-based system engineering; FRAM





## 1. Introduction

*The Problem*

**"All models are wrong, but some are useful" (Box and Draper (1987))**

Within our world, our technological, physical, social and political environments are becoming increasingly complex and less predictable (and intelligible?). "Artificial" enhancements seem actually to make them more opaque and less easy to understand or predict their behaviours. So, there is a growing sense that perhaps we are not as in control of our lives as we should be. To those of us charged with ensuring our systems can be operated safely, this is a very real challenge. How do we make the case that the systems we design, install, and operate are "safe"?

To do this, we need to convince ourselves that we understand and can predict how they will behave in real world situations. We prefer to do this by modelling the systems to discern the inner workings of the black boxes. But are our models adequate for the task, because to paraphrase Lord Kelvin[1], if we can't model it, we can't manage it!

The objective of the paper is to describe the development of a new, formal approach to modelling complex systems, based on a concept developed by Hollnagel, of modelling complex systems as dynamic collections of interacting, interdependent functions. These are (abstracted) entities which produce the intended effects of system components, or component combinations and are agent agnostic. Hollnagel called it a Functional Resonance Analysis Method (FRAM), to emphasise the dynamic effects possible in these interacting, interdependent systems of functions.

This further development into a bona fide system model is set out here against a background of the problems with current practice in the light of the increasingly complex systems we are dealing with, to underline our present limitations. It identifies the need and criteria for a better methodology and sets out the basis and applicability of these new FRAM solutions to this problem, and outlines their derivation, and applications, for scrutiny. In a similar vein, the current software solution, is now publicly available on the GitHub website for community usage, appraisal, and feedback under a Free Software Foundation general public license.

Thus, it sets out what we could be using now, and how we need to further develop this, or other more advanced approaches, to ensure our modelling competencies for the future.

## 2. Current Modelling Approaches

### 2.1. The Formal Basis of a Model

Models are ubiquitous in the scientific literature as well as in other places. Most researchers tend to propose models as part of their work. But what is a model, actually? The main features of what we tend to call models are a set of components, or entities that are connected by lines. The model components can be enclosed in boxes, or other geometrical shapes, or they can simply be names. The components are connected by lines that usually have a direction, and sometimes these point in both directions. As Coombs [1] points out:

*"The general purpose of a model is to represent a selected set of characteristics or interdependencies of something, which we can call the target or object system. The basic defining characteristics of all models is the representation of some aspects of the world by a more abstract system. In applying a model, the investigator identifies objects and relations in the world with some elements and relations in the formal system".*

It is therefore necessary that the model is based on some sort of formal system that can be used to express or represent the "objects and their relationships in the world" that are being investigated. It is not enough to know what you want to study or represent. It is also necessary to have a concise way of doing it. The formal system which provides the basis for the model, therefore, represents the assumptions about "the world", about what lies behind the phenomena being studied or analysed. Without a formal system, the model in practice becomes meaningless.

Business System Management

Some examples of these formal system models, include Business (System) "Intelligence" approaches, such as are supported by commercial software such as Microsoft Power BI. These are very powerful ways of organising detailed information on component parts of a system and their interconnections.

This approach has been formalised by designers of software systems into an extensive library of "models" that can be utilised in Model Based System Engineering [2]

*"Model-based systems engineering is the formalized application of modelling to support system requirements, design, analysis, verification and validation activities beginning in the conceptual design phase and continuing throughout development and later life cycle phases".*[2]

Other modelling methodologies for business management include the Business Process Modelling approach [3] which outlines a formal process for visualising a sequence of tasks needed to carry out specific processes.

Many of the models prescribed in the Model-Based System Engineering Approach, have been developed to support the better design of software systems, based on an object-oriented design approach [4] This, as the name implies, focuses on the decomposition of systems and mapping relationships between the objects (components) identified. Another earlier approach, however, which seems to have fallen out of favour lately, looked at software systems as a system of interacting subsystems, which they called Funktions [5], from which they developed a structure of interacting subsystems, which were abstractions

of the details of the activities and entities needed to fulfil the Funktion's intended task. This Structured Analysis and Design Technique (SADT) is very useful in tracing what is happening in complex software packages.

### 2.2. Safety Management

Most current approaches to modelling systems to manage their "safety" are based on finding the faults, or defects in design, which could compromise the reliability, and hence the safety, of the system. They rely on traditional cause-and-effect analyses using methodologies which involve tracing the effects of individual component "failures" (Failure Modes and Effects Analysis [6], to construct Boolean logic trees, such as bottom-up "Fault" Tree analysis, or top down "Root Cause" analysis [7].

The principle behind these approaches is based on the prior decomposition of the systems into simpler pieces, or component parts, whose behaviours can be more easily established and predicted (decomposition until behaviours can be modelled deterministically and safe behaviours predicted confidently!).

Thus, this is predominately a defensive approach, which accepts the complexity and attempts to be able to contain excursions from (ab?) "normal" behaviours.

Further extensions of this approach recognise potential abnormalities which could be caused by external hazards, or internal errors by operators, or management, by attempting to ensure that there are no holes in the layers of their safety management systems [8], or by ensuring there are sufficient preventative barriers and/or adequate mitigating measures, to ensure any system excursions are survivable. (This is often illustrated as rows of dominoes or multiple barriers in "Bow Ties") [9].

With more complex systems, these barriers themselves can be complicated systems, which inevitably add layers of complexity to the system. It is then a question of trying to assure designers and operators that these more sophisticated "barriers" (Safety Critical Systems), can actually make the total systems safer, not add additional "failure" modes and probabilities [10].

So, the credibility of the safety cases made to justify the acceptable operation of these systems depends on the confidence we have in the ability of external "barriers" to address events which we know can be emergent and unexpected in complex systems in the real world.

There is thus a question mark over the ability of the current approaches to credibly and responsibly assure the safety of complex systems. Detailed knowledge of the pieces, components and subsystems is a crucial prerequisite, but is it enough without an understanding of how the whole system will react?

### 2.3. Real World Systems

The high visibility and social impact of actual failures in the operation of complex systems such as public and industrial installations and processes has encouraged safety professionals to propose "models" to explain the causes as not merely component unreliabilities. Rasmussen [11], indeed, discusses "risk management in a dynamic society; a modelling problem". Perrow [12] identified the problem as the tight couplings (between what?) involved in complex systems. Rasmussen proposed to model these systems as a hierarchy of entities, (Accimaps), influencing how operations at the sharp end are carried out.

Hopkins [13] (has used these Accimap models very successfully to explain how a number of high-profile incidents have occurred.

Leveson [14], after the Columbia space shuttle incidents, formalised these hierarchies as "control loops" designed to regulate these complex systems as conventional engineering diagrams; the so-called Systems Theoretic approach (STPA).

But, as Ackoff points out, although these models map out the components and their place in the system and hierarchy, they do not address the fundamental issues arising in

complex sociotechnical systems and accidents, such as the nature of the couplings, (Perrow) and the dynamic evolution and emergence of behaviours (Rasmussen).

What is needed, and is lacking in most of the current modelling approaches, is a way of exploring and understanding how these couplings and interactions result in the observed behaviours of complex systems in the real word. The Rasmussen and STPA abstraction and hierarchical arrangement of entities, not components, is a start, but we need a more formal method to explore much more deeply the effects of these couplings and the dynamic nature of their interactions. The exceptions to this appear to be the Structured Analysis Design Technique and the Functional Resonance Analysis Method.

## 3. The Functional Resonance Analysis Method

### 3.1. A Solution?

The Functional Resonance Analysis Method (Hollnagel [15]) does refer to a formal methodology, based specifically on four principles (equivalence of successes and failures, approximate adjustments, unpredicted emergence, and stochastic resonance). More specifically the FRAM—qua method—clearly specifies the model entities and their possible relationships. The purpose of the FRAM is to develop a description, in the form of a model, of the functions that are needed—or have been used—to carry out an activity, or a task, and to show how they mutually depend on each other in various ways (also known as coupling).

In some respects, the FRAM could be seen as a logical development of the SADT approach. The "components" must be functions—something that is done, has been done, or can be done either as background functions, or foreground functions. And the relationships must be described as one of the six aspects (instead of the SADT's set of four), that are defined for the functions, as shown in the diagram below (Figure 1).

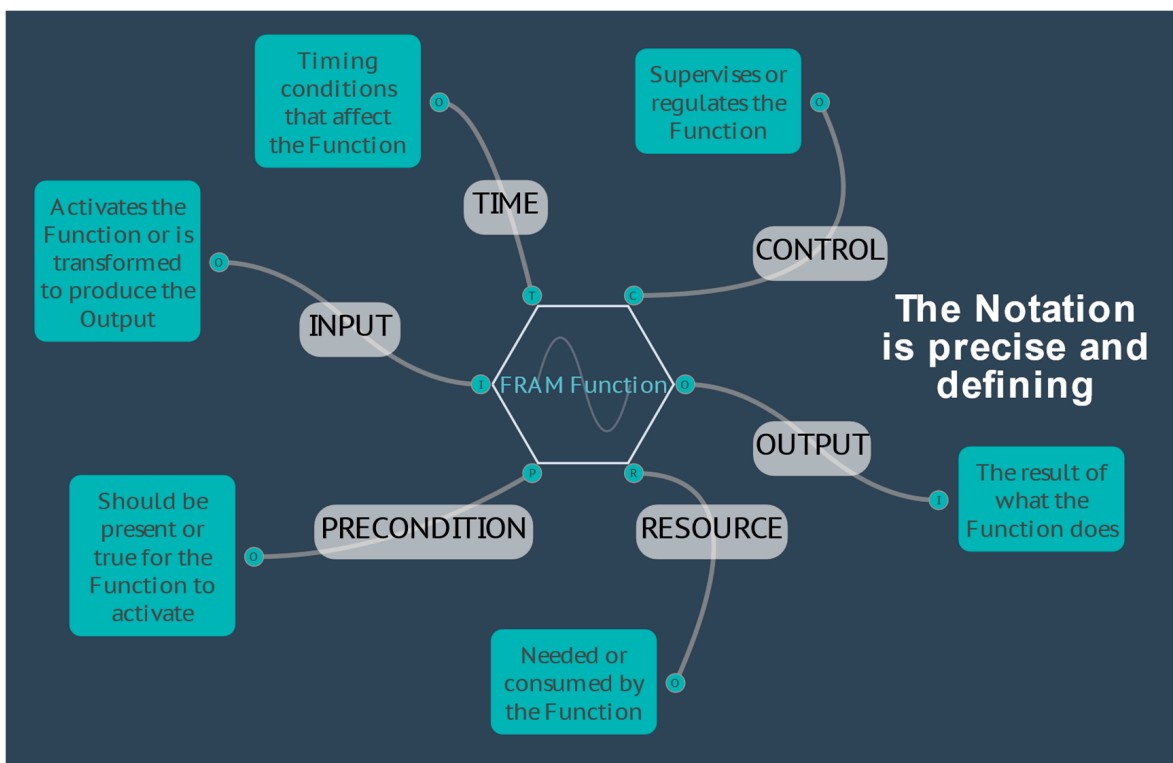

**Figure 1.** The FRAM Function.

The relationships between functions are furthermore defined by means of the upstream–downstream couplings. Because of this, a FRAM model can be seen as a kind

of program—as a series of instructions to control the way in which the model operates or "performs".

(For the remainder of this document, 'Function' with a capital 'F' will be used when referring specifically to a FRAM Function in the context of the FRAM method or a resultant FRAM model.)

The FRAM Model Visualiser

In their initial application, FRAM models were developed by hand as linked hexagons, which was very time consuming and was a disincentive to their wider adoption. After many attempts, a software solution, the FRAM Model Visualiser (FMV), was developed by Hill [16] and transformed the uptake of the approach, as more users were able to spend their time in analysing, rather than painstakingly, manually creating the models. It is now an open-source code, with user support and documentation on a GitHub site that is available for open access.[3]

The further development of the FMV software has considered user feedback and publications to expand functionality that has supported the community of users and further research and development of FRAM.[4] The evolution of the sophistication of the software capabilities became a continuous feedback loop to stimulate researchers to explore further the potential capabilities of the method.

### 3.2. Current Applications of the Method

The objective of this paper is to enable and point towards the future applications now possible with the methodology. This is not to ignore the many applications that have been undertaken and published in the intervening years. For reviews of this work, there are several excellent publications which set out how various people have applied the methodology to a wide range of situations. Perhaps two of the best are Patriarca [17] and Smith [18].

These show applications of the methodology, from construction-worker safety to nuclear power operations. The reviews show the main activity has been in applying it to study intense sociotechnical systems, such as in aviation and healthcare.

A series of videos sets out well the current applications of the methodology.[5]

### 3.3. The Further Development of the Methodology

3.3.1. Variability

As well as FRAM's SAFETY II principle of the equivalence of success and failure, the approach also emphasises the principles of Approximate adjustments, Emergence, and Resonance. Each of these involves the propagation of variability in the interactions of the Functions in a system to give nonlinear, non-predetermined outcomes. The problems of tracking this variability have always been a challenge.

The FMV provided the ability to name a Function and enter a description, as suggested by the original FRAM method, but it soon become apparent that users required more. In the original Hollnagel book, Functions were characterised into three types, consistent with the widely used distinction in describing complex socio-technical systems, i.e., technological functions, human functions, and organisational functions. The variability of each type was then considered further, and the classification into phenotypes was used to describe the manifestations of their performance variability. A simple solution was explained in detail [19], using two phenotypes—variability with regard to time and variability with regard to precision.

Early versions of the FMV had this ability to set the type of a Function to technical, human or organisational, and to assign a qualitative variability with regard to the two phenotypes as described by the simple solution in FRAM. This variability was then displayed as wavy lines on the appropriate Functions in the visualiser. An early attempt at visualising variability qualitatively was the FMV Variability Tracer, which took the variability assigned using the two simple phenotypes and showed the propagation of the variability across

potential couplings, displaying it as coloured bars on the Functions. Without a structured approach as to how the potential couplings became actual couplings during an instantiation of the model, the result was simplistic, and it did not attempt to show the results quantitatively. However, it was useful in illustrating trade-offs between the two phenotypes using Hollnagel's ETTO principle, where, if a Function already displayed variability but was forced to be 'on-time', then the result was 'less precise', or conversely, if it was forced to be 'precise' then it was more likely to be 'late'.

A more elaborate solution was also proposed in the original Hollnagel book [16] using eight phenotypes taken from failure modes that are part of most safety models: speed, distance, sequence, object, force, duration, direction, and timing. The FRAM analysis however, considered these as describing variability, not failure. Users were also developing their own libraries of phenotypes and attaching other information and data to Functions to explain them in more detail and to consider how they interact with other Functions, with some users defining their own sets of phenotypes specific to their use (see Franca [20]).

Because the FRAM method was always intended to have this level of flexibility so that it could be applied to many different applications, it was decided not to further develop the hard coding of phenotypes into the FMV software. Development instead focused on creating a Metadata layer for Functions, so that users could define and attach their own data for phenotypes, descriptions, or any qualitative information specific to individual Functions.

The FMV was by now widely accepted as the standard for creating and visualising FRAM models, but many researchers were using their own methods to perform quantitative analysis. Three notable examples are outlined below.

- MyFRAM was developed by Patriarca [21] and used as a powerful pre- and post-processor by various researchers, where an FMV data file could be read into the application and the data could be viewed and edited in spreadsheet views, including calculations and other qualitative or quantitative analysis of variabilities to predict resonances. The resulting model could be output as an FMV data file to be displayed again in the FMV software (for example, Patriarca's Monte Carlo application).
- Early on in the FMV development, a successful integration was made (Slater [22]) to pass data to an external Bayesian Belief Network (BBN) processor, using the open-source i-Depend software, returning results to the FMV for visualisation.
- Another example of quantitative post-processing employed a Fuzzy Logic approach (Takayuki [23]), which manipulated FMV Functional properties in an external program, creating an output that could be displayed and visualised in the FMV application. Variability was represented by a Degree of Variability (DoV) property of a Function, modified by couplings with other Functions.

### 3.3.2. Validation

There was an interest in the early stages of the development of the FRAM methodology in somehow "validating" or checking that the "mind map" models were internally consistent, as well as satisfying users that they were a realistic representation of the systems involved.

Accordingly, Hollnagel [24] produced the FRAM Model Interpreter (FMI), which recognised that the internal method of the Function's action was, in effect, equivalent to what is often specified as a line of computer code:

For example—(IF: aspects present THEN: execute the Function ELSE: Wait).

The FMI then executed this "program" and flagged any anomalies, inconsistencies, or logical errors in a log for correction.

From a programming perspective, the FMI has been developed as a production system (sometimes called a production rule system). Production systems were widely used in artificial intelligence in the 1980s and are defined as follows (Selfridge, [25]):

*A production system (or production rule system) is a computer program typically used to provide some form of artificial intelligence, which consists primarily of a set of rules*

*about behaviour, but it also includes the mechanism necessary to follow those rules as the system responds to states of the world.*

This was a much-needed check, but it was a standalone piece of software into which FRAM model files had to be imported, and the resulting analysis and results were text-based.

### 3.3.3. Purposes of the FMI

- One purpose of the FMI is to check whether the model was syntactically correct.
  - An important part of that is the identification of "orphans" (i.e., unconnected aspects), which the FMV already provides as a visual check. To begin the FMI interpretation, all defined aspects must connect to other functions as potential couplings.
  - Other problems that the FMI flags as invalid are potential auto-loops, where the Output from a Function is used directly by the Function itself. The FMV, however, will not make this type of connection and flags it as an orphan.
  - Finally, there is a check on the Inputs and Outputs by FMI Function Type. If a Function has aspects apart from Inputs and Outputs, then it is, by definition, a foreground Function and it must have at least one Input and one Output.

- A second purpose of the FMI was to understand the difference between potential and actual couplings among Functions. A FRAM model describes the potential couplings among Functions, i.e., all the possible ways Functions are related as specified by their aspects. In contrast to that, the actual couplings are the upstream–downstream relations that occur when an activity is carried out, which means when the FRAM model is realised for a set of specified conditions (a so-called instantiation).

- A third purpose was to determine whether the activity described by the model will, in fact, develop in practice. In a FRAM model, each foreground Function defines a set of upstream–downstream relations through its aspects. The question is whether these relations are mutually consistent and whether they will, in fact, allow an event to develop as intended.

- It is all too easy in a complicated model to have Functions that mutually depend on each other, which in practice may lead to conditions where Functions wait forever for an aspect to become fulfilled. The FMI can identify these cases by interpreting the model step-by-step while keeping track of the status of all the aspects and activation conditions.

- A further purpose is to investigate the consequences of variability of Functions. In the Basic FMI this is carried out indirectly, by specifying the conditions under which a Function may become activated, rather than by considering the variability of outputs directly.

### 3.3.4. Metadata

Hill, who had developed the original FRAM Model Visualizer (FMV) software which was, by then, universally employed to build the models, then combined this "interpreter" facility into the FMV software to give a platform which could now not only validate the structure and executability of the models, but display the results visually within the model view. This made it much easier to validate and troubleshoot deficiencies in a model, while also providing a way to present visually specific instantiations of a model, where some of the potential couplings between aspects had become actual couplings.

This version of the FMV software, with integrated FMI, has been made available as free software under the terms of the GNU Affero General Public License as published by the Free Software Foundation.

With the FMI providing a formal structure for the interpretation of a model, including how a Function would consider its aspects coupled with upstream Functions before it would produce an output, the door was opened to more structured forms of quantitative analysis.

Hill had earlier introduced the concept of "Metadata" into the FMV, allowing a user to define their own data attached to a Function. By defining a Function's metadata as a

potentially unlimited list of user-defined key–value pairs, these could be used for various purposes, such as descriptive information, properties, and phenotype definitions, as well as being able to hold quantitative data for consideration by analytical methods.

This now replaced the need for the hard coding of fixed data types in the software, such as the Function description and the simple solution previously adopted of phenotypes and variability.

Hill then added an equation parser to the metadata, where the value of a key–value pair could be calculated by a simple equation during the FMI interpretation, instead of holding a fixed value. This variable value would then be available to downstream Functions through aspect couplings to be considered in other Function calculations of metadata, thus propagating calculated data throughout the model.

While not being directive, this allowed the user to define their own method of calculation and quantification, where metadata could be used, for example, to quantify changing properties of Functions under varying conditions, or to give indications of variability numerically, such as fuzzy-logic degrees of variability or normal distributions. This allowed real "resonance", both positive and negative, to be picked up.

3.3.5. Dynamics

The FMI uses 'Cycles' to track the propagation of actual couplings during an interpretation, beginning with the Entry Function(s) and progressively cycling through all Functions that are activated when they receive an Input from an upstream Function, until the end conditions are met, where all Exit Function(s) have activated. During this cycling, when a Function receives an Input, it considers all of its aspects before it produces its Output(s). By default, if ALL aspects are present (they have become an actual coupling with an upstream Function on a previous cycle) then the Output is produced. This can be adjusted using an FMI Profile on each aspect type, on each Function, where ALL can be replaced by ANY or NONE.

Using the FMI-cycle logic and profile conditions as the basis, Hill added the calculation of metadata equations to this process, and extended other features of the FMI.

- The activation of the Entry Function(s) on the initial Cycle 0 of the FMI can be repeated at a set cycle interval, effectively pulsing the model.
- The FMI interpretation can be extended beyond the default stop conditions (when all Exit Function(s) have activated) for a set number of cycles, or continuously until manually stopped.
- An equation can be added to the FMI Profile to consider the aspects and their metadata in more detail, before an Output is produced, potentially putting a Function on 'Hold' until conditions are met on a subsequent cycle.
- A table of lookup values can be referenced by the equations.

The combination of these extensions enabled the analyst to vary the choreography of the way the system propagated, in terms of the cycling and sequences, and the ability of Functions to "Hold" until the upstream aspects met user-defined criteria, which were more flexible than the standard FMI 'profile' conditions. This solved a vexing problem for analysts, where there was a choice of routes optional in the development of an instantiation (Plan A vs. Plan B). This also allowed the tracking of emergent behaviours as various options interacted.

This cycling facility then opened the way to the debates about real time, ordinal time and cycle sequence; the multiple iterations in cycling made a Monte Carlo approach to predicting outcomes a possibility, where instantiations could be repeated programmatically with random, or distributed changes in initial data.

This availability of a way to represent formally the properties of the inputs and a mathematical description of the Function's method of execution, makes this approach potentially very powerful.

## 4. How It Works

### 4.1. Detailed Description of Metadata in FRAM

A FRAM Function represents an activity or process that changes something in the system, and this change can have an effect on any other Function in the system with which it interacts. The metadata facility sets out to allow us to see the effect of these changes and to follow the consequences of these events.

Consider a Function has a property that can be modified after its interaction with another Function. Call this property/parameter a "value" and identify it with a "key".

The FMV allows us to specify these key–value pairs as metadata (Figure 2).

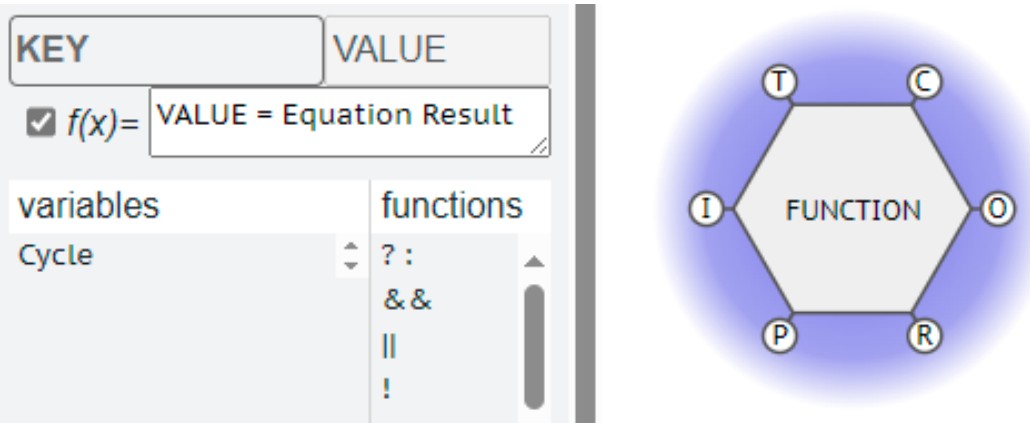

**Figure 2.** Metadata key–value pair.

In this way, the metadata can be defined for different types of data, including the following:

- Fixed constants that represent properties of a specific Function, like a name, description, or other properties.
- Variable properties of a specific Function that can be modified using equations based on other metadata available from upstream Functions.
- Global constants that are set by the starting Function and passed on through the couplings.
- Global variables that are passed on through the couplings but can be modified by Functions using equations as they pass through.

Consider the Function "to boil water", using an electric kettle.

One of the parameters of critical interest would be the TEMPERATURE of the water being boiled.

There would then need to be a KEY label for this Temperature, which would have a VALUE calculated from an EQUATION relating the power supply wattage, the quantity of water in the kettle and the time allowed for the kettle to boil.

These extra parameters, POWER, QUANTITY and TIME, can be VALUES for these new KEYS, transmitted by the interaction with upstream, or background Functions. So, the FMV does not just check that these upstream aspects are present, it can also read and process the information from the interaction as a measurable (calculable) effect on the Function's OUPUT.

The Temperature of the water produced by that Function could thus have a calculated VALUE as a mathematical function of the VALUE of the strength of the power supplied, the VALUE of the quantity of water, and the VALUE of the time that the kettle is allowed to take. The equation is shown in Figure 3 and the FRAM model min Figure 4.

Further, the TEMPERATURE of this water could have a direct effect on the QUALITY of the tea produced, if this "boiled water" OUTPUT is an ASPECT for a Function "To pour on to tea leaves". So, we can have a KEY–VALUE pair for the quality of the tea produced by this Function. This can then be calculated from the TEMPERATURE of the water received and perhaps a TIME VALUE from an upstream Function "To control the time it is allowed to stew".

In a conventional FRAM analysis, the possible effects of cooler water or shorter times could be flagged as a variability to be noted. With the metadata we can now follow/predict the development and the quantitative effect of the ripples produced by this variability, right throughout the instantiation, and track any "emergence" of unexpected effects, or "resonances".

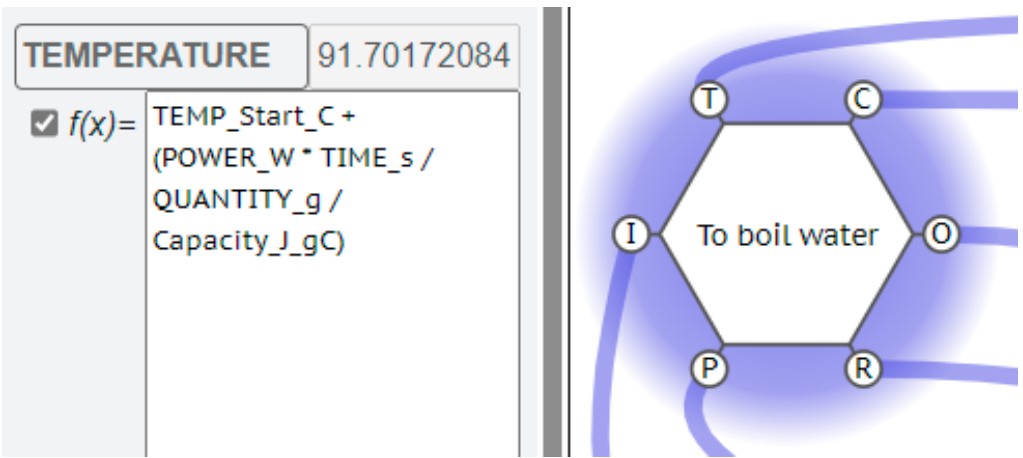

**Figure 3.** TEMPERATURE equation on Function "To boil water".

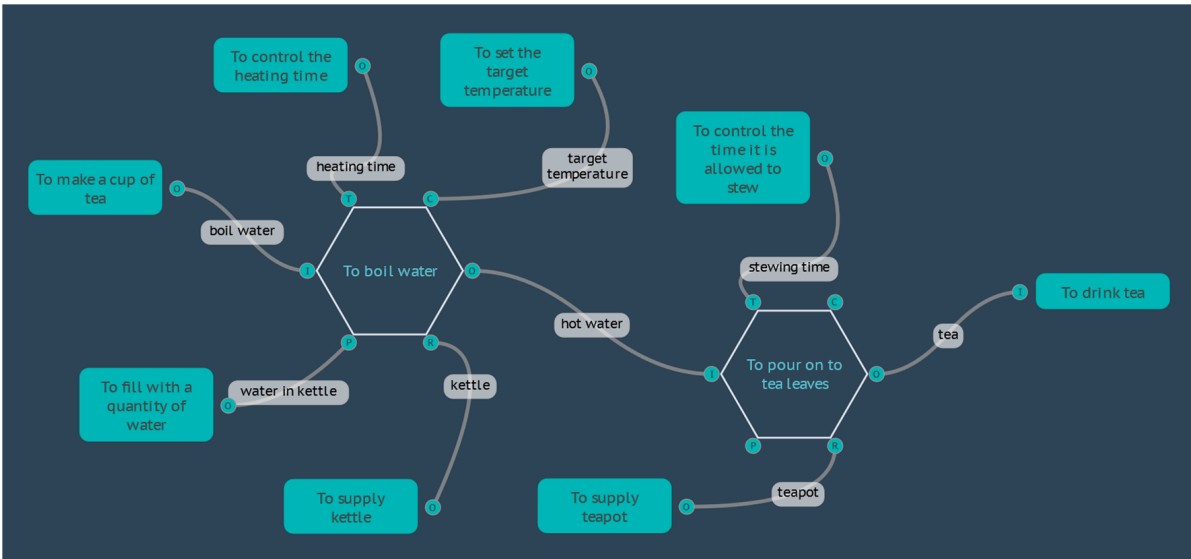

**Figure 4.** The "Boil Water" FRAM model.

Because the metadata is defined by the user, the FMV does not direct any particular method of analysis, but rather, by maintaining an open configuration, allows for the possible use of many different forms of interpretation and analysis. This is in keeping with the purpose of the FRAM, which is a method to build a model of how things happen, not to interpret what happens in the terms of a pre-existing model. By combining FRAM with the FMV and Metadata, it should be possible to model any complex system and define a meaningful analysis.

### 4.2. Examples of Metadata

#### 4.2.1. Boil Water

If we take the Boil Water model used in the previous section, taking the equations as they are and setting aside any discussion about the physics and assumptions made, we can examine this as an example of how the metadata can be used to calculate and display results, and how the model can be used to simulate the system under varying conditions.

The formula used to calculate the temperature of the water after a given heating time is based on a simple specific heat formula, shown here as it is represented in the FMV using somewhat descriptive key names:

$$\text{TEMPERATURE} = \text{TEMP\_Start\_C} + (\text{POWER\_W} \times \text{TIME\_s}/\text{QUANTITY\_g}/\text{Capacity\_J\_gC})$$

where

- TEMP_Start_C is the starting temperature of the water in degrees Celsius (°C);
- POWER_W is the power rating of the electric kettle in Watts (W), which is the same as Joules per second ($J \cdot s^{-1}$);
- TIME_s is the time that the water is allowed to heat, in seconds (s);
- QUANTITY_g is the quantity of the water that is being heated, in grams (g);
- Capacity_J_gC is the specific heat capacity of water, at 4.184 Joules per gram, per degree Celsius ($J \cdot g^{-1} \cdot {}^{\circ}C^{-1}$).

The calculation is entered into the Metadata for the Function <To boil water>. The data used in the calculation is 'supplied' by other background Functions, initially as constants that represent properties of the water, kettle, and time, as shown in the following Table 1.

**Table 1.** Initial key–value pairs for the "Boil Water" model.

| Function | Key | Value |
|---|---|---|
| <To fill with a quantity of water> | TEMP_Start_C | 20 |
| | QUANTITY_g | 1000 |
| | Capacity_J_gC | 4.184 |
| <To supply kettle> | POWER_W | 1000 |
| <To control the heating time> | TIME_s | 300 |

When the FMI interpretation is advanced, on Cycle 0, the background Functions supply these data through their couplings to the aspects of the <To boil water> Function. The Entry Function <To make a cup of tea> also activates and supplies the need for boiled water through the Output [boil water], which couples with the Input of the <To boil water> Function. On the next FMI Cycle, 1, the conditions for the activation of <To boil water> have been fulfilled, such that the aspects are all present and an Input has been received. The TEMPERATURE equation is calculated (91.7 °C) and the Output [hot water] is created.

At this point we can display the results of the TEMPERATURE equation by activating the display results panel that allows for the selection of a Metadata key, with the corresponding value displayed as a configurable colour scale (Figure 5). Advancing the FMI interpretation to Cycle 2, the next downstream Function activates <To pour onto tea leaves>.

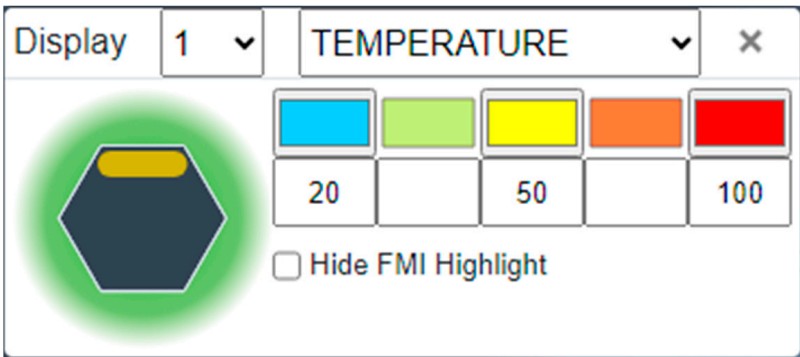

**Figure 5.** The FMV Display results setup, showing the colours and ranges for strength of tea.

To complete the example model, this function has an equation that takes the temperature value supplied by the [hot water] Input together with a [stewing time] constant (3 min) to calculate the quality of the tea produced, where the result is either "Weak", "Good", or "Strong". These results can also be displayed against a second colour scale.

The FMI interpretation concludes on the next Cycle, 3, with the Exit Function <To drink tea>.

The results of the completed interpretation are displayed in the FMV as shown in Figure 6, with a water temperature of 91.7 °C and a tea quality of "Good" displayed on their Functions as a red bar and green bar, respectively.

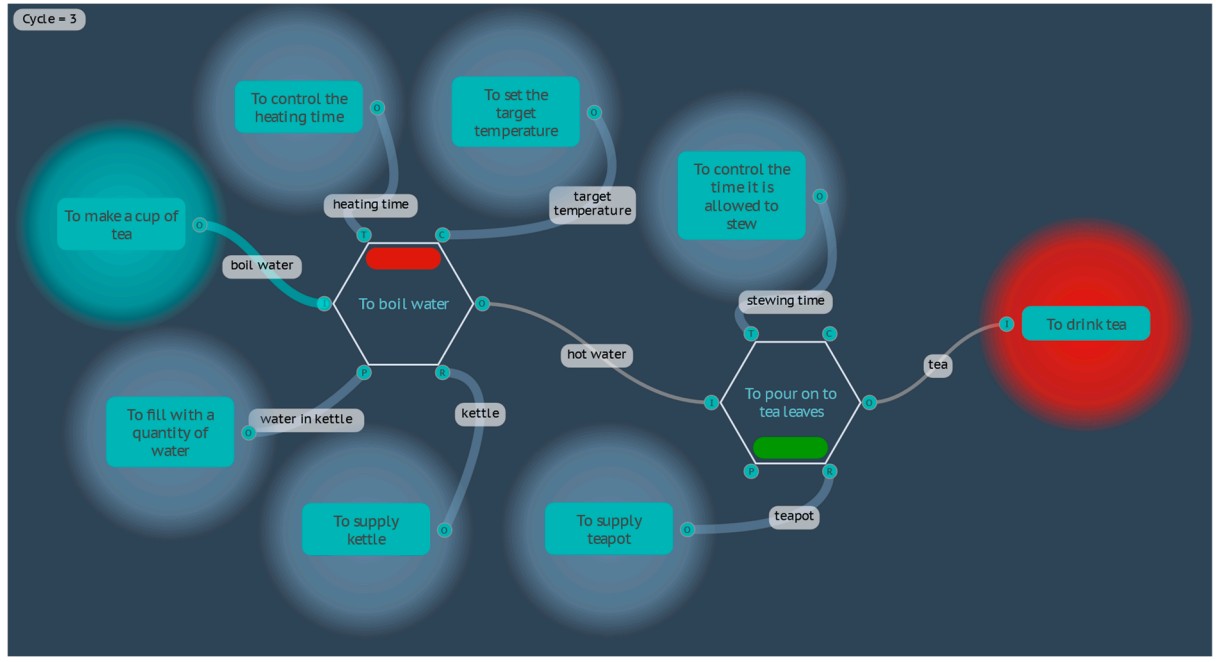

**Figure 6.** Initial FMI display results for the "Boil Water" model.

The results can also be output as a data table, as shown in Table 2.

**Table 2.** Initial FMI results for the "Boil Water" model.

| Cycle | Function | TEMPERATURE | QUALITY | TIME_stew_m | POWER_W | QUANTITY_g | TEMP_Start_C | Capacity_J_gC | TIME_s |
|---|---|---|---|---|---|---|---|---|---|
| 0 | To make a cup of tea | | | | | | | | |
| 0 | To control the time it is allowed to stew | | | 3 | | | | | |
| 0 | To supply kettle | | | | 1000 | | | | |
| 0 | To fill with a quantity of water | | | | | 1000 | 20 | 4.184 | |
| 0 | To control the heating time | | | | | | | | 300 |
| 0 | To set the target temperature | | | | | | | | |
| 0 | To supply teapot | | | | | | | | |
| 1 | To boil water | 91.70 | | | | | | | |
| 2 | To pour onto tea leaves | | Good | | | | | | |
| 3 | To drink tea | | | | | | | | |

Now that the model is able to simulate a set of specific conditions, an instantiation, we can now vary the initial data to create different scenarios for alternative instantiations of the model.

For the second instantiation, let us double the amount of water supplied in the kettle, where QUANTITY_g now equals 2000 g (about 2 L). If the heating time is still constrained to five minutes (TIME_s equals 300 s), then executing the model yields a water temperature of 55.85 °C and a quality of tea that is "Weak", shown in Figure 7 by the orange and pale-yellow results bars, respectively.

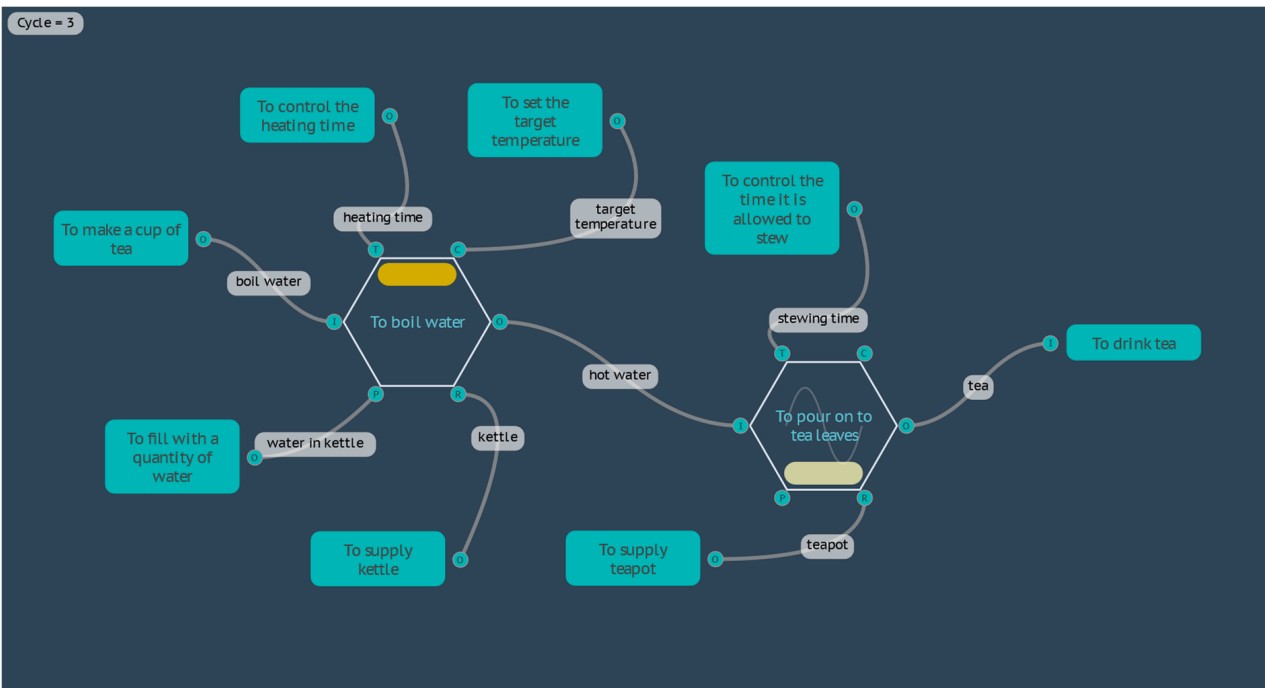

**Figure 7.** Results for the "Boil Water" second instantiation.

For a third instantiation, let us return the water quantity to 1000 g, but increase the stewing time on the Function <To control the time it is allowed to stew>, from 3 min to 7 min. Now the results return a water temperature the same as the first instantiation of 91.7 °C, but the quality of the tea is now "Strong", as shown by the brown results bar in Figure 8.

This modelling could be used in a situation where the total time to boil the water, allow the tea to stew, and then enjoy drinking the tea, is constrained: but other parameters such as the quantity of water and the electrical power of the kettle could be optimised to produce quality tea.

One of the background Functions, <To set the target temperature>, has not been used so far in this example. This is provided as an alternative Control aspect where the time is unconstrained, and the kettle would be allowed to continue heating until a set temperature has been achieved. In this case, the formula is rearranged to calculate the time required, where

$$\text{TIME} = (\text{TEMP\_target\_C} - \text{TEMP\_Start\_C})/\text{POWER\_W} \times \text{QUANTITY\_g} \times \text{Capacity\_J\_gC}/60$$

Given the starting values of the initial instantiation, the time required to reach 100 °C would be 5.58 min. This could then be passed on to downstream Functions for other calculations, such as the time remaining to enjoy drinking the tea before the break time is over. An application of this type of time calculation is outlined in the Formula 1 Pit Stop model in Section 4.3.

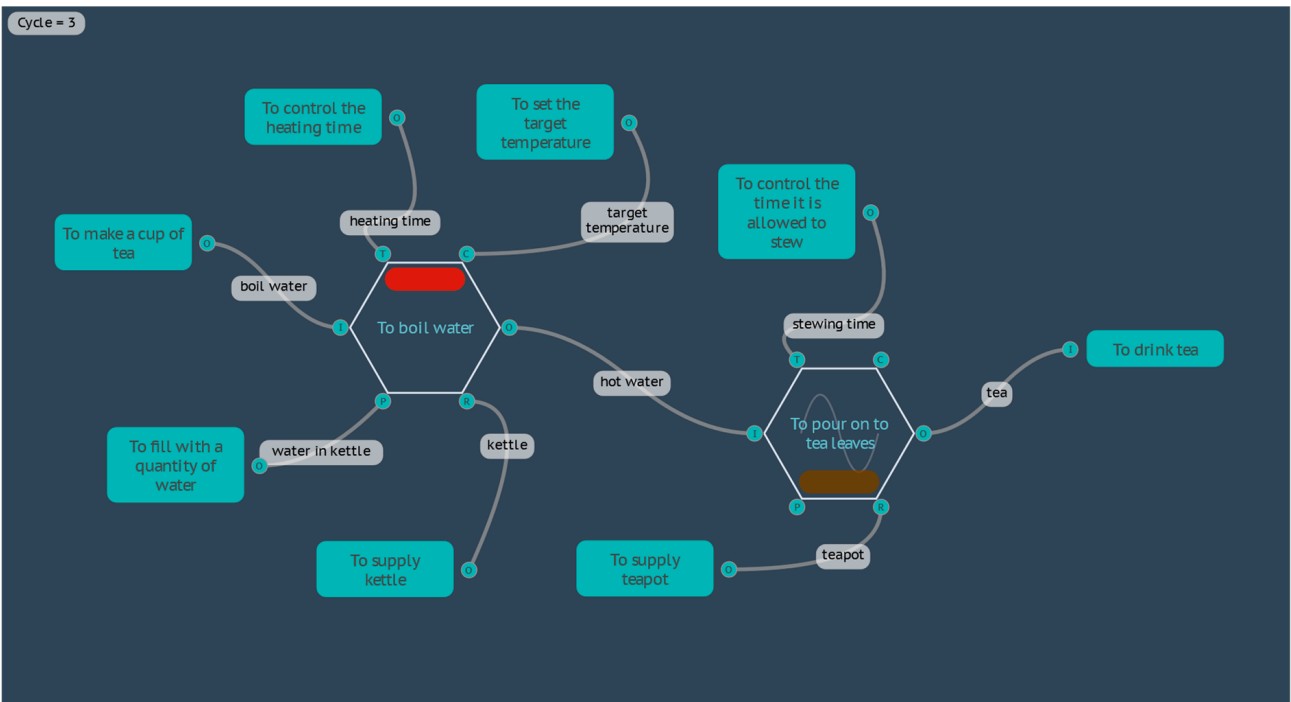

**Figure 8.** Results for the "Boil Water" third instantiation.

It should be noted that the starting temperature of the water in the kettle is another factor that could introduce variability into the results, but which so far has been set as a constant, at 20 °C. As already demonstrated for other parameters, we could manually change this value for subsequent instantiations to model the effects of a range of different values.

Another option for modelling this type of variation in start parameters is to set the model to cycle repeatedly but with different Metadata values. The model takes 3 FMI Cycles to progress from the Entry Function to the Exit Function, so if we set the Entry Function to activate every third Cycle, then the model will restart every time it completes a full interpretation. We also set the FMI Control to ignore the default stop conditions and run continuously.

Next, we have five different options for varying the starting temperature of the water automatically for each instantiation:

1. For the Metadata key TEMP_Start_C on the Function <To fill with a quantity of water>, the value can be changed to 0 to set the starting point for the first instantiation, and then an equation added, where TEMP_Start_C = TEMP_Start_C + 1. This will start with 0 °C and increase the temperature by one degree every instantiation.

2. The equation can be set to a pseudo-random number within a given range. For example, to start an instantiation with a random integer between 10 and 30 (inclusive), the equation would be TEMP_Start_C = rndi(10,31). Double-precision floating-point numbers can also be generated using rnd().

3. The equation can be set to return a pseudo-random number which, on repetition, approximates a normal distribution. For example, a normal distribution with a mean of 20 and a standard deviation of 10 would be TEMP_Start_C = rnorm(20,10).

4. A table of values can be imported into the model, and an equation used to return a value from the table, such as lookup (row number, column number). In our example, the row number can be set from the available system variable that contains the current Cycle number, Cycle, and the first column of the data table, such that TEMP_Start_C = lookup(Cycle/3,0).

5. Similar to the previous case, a value can be referenced from a data table, but using a cross reference between a row name contained in the first column, and returning data from another column referenced by a column-header name, such as

TEMP_Start_C = xref(Cycle,"Temp"). In this case, the data table does not need to be sorted or sequential.

After allowing the interpretations to run for the desired number of Cycles and repetitions, the results table can be interrogated (and exported) to analyse the results, for example, to find the minimum starting temperature that would produce quality tea within the given time constraint (this happens at about 19 °C, and at Cycle 56, using option 1).

This example has been kept intentionally limited, with only two foreground Functions, to the point where the analysis conclusion above could be calculated directly without modelling. However, the intention is to demonstrate the basics of modelling that can be extended to more complex models. Even given the premise in this example, the boundaries of the model could be expanded so that the background Functions themselves become foreground Functions and the parameters discussed are influenced by other upstream Functions. In this way, we could begin to model variability that is influenced by numerous interrelated couplings with the possibility of discovering unexpected emergent results.

### 4.2.2. Plan A or Plan B

It is not uncommon in FRAM models for a Function to have more than one Output that may collectively couple with a single downstream Function, or the different Outputs may couple with different downstream Functions. The realisation of these potential couplings into actual couplings during an instantiation could differ based on the conditions of the specific instantiation, in effect causing conditional branching of the model during the instantiation.

In the basic FMI, a Function that receives an Input then considers the presence of its aspects according to its defined FMI profile before activating and producing its Outputs. If it activates, then all Outputs are created and become actual couplings with downstream Functions. If it does not activate, then no Outputs are created. This basic default setup does not consider the Metadata available from upstream couplings before making the 'decision' to activate, and if it does activate it cannot choose to create only one Output over another.

If the interpretation analysis of a model requires this more complex functionality of conditional activation and branching, then it is extended through the use of a special Metadata key named Activate. To illustrate, consider the following "Plan A or Plan B" model which, under default conditions, will progress the FMI interpretation to the stop conditions with the activation of two Exit Functions, <To carry out Plan A> and <To carry out Plan B> (Figure 9).

First consider the Function <To meet the critical conditions>. To model an instantiation where the Output of this Function is never realised, and does not create an actual coupling with the downstream Function, then we can add the special Metadata key Activate with a value of 0. If the value is 0 or false, then the Function will not activate (Figure 10).

When the FMI interpretation is run, it stops at Cycle 2 with no Functions activated. The Function <To meet the critical conditions> is on "hold" because of the Activate key, as shown by the yellow highlight in Figure 11. The downstream Function <To execute the critical function> does not activate because the conditions of the default FMI profile for this Function are not satisfied, in that the precondition aspect is not present.

For the next illustration, we will set this Activate key to a value of 1, which means that all Outputs will be created. This has the same result as the initial model with the activation of the two Exit Functions.

Next, take the other background Function <To provide the required materials> and create a Metadata Key named Materials, with a value of 100.

Now consider the Function <To execute the critical function>. We can model an instantiation where this Function looks at the materials available and makes a decision on which Output to create. For example, if the Materials are greater than or equal to 70 then [Execute Plan A], otherwise [Execute Plan B]. This is achieved by adding an Activate key to the Function with a conditional formula. If the value of an Activate key (either fixed or calculated) is any other text that is not 0, 1, or false, then it looks for an Output name that is

contained within the text, and activates that Output specifically. For this example we enter the formula

$$\text{Activate} = \text{R4\_Materials} \geqslant 70\ ?\ \text{"Execute Plan A":"Execute Plan B"}$$

The first part of the formula up to the question mark is evaluating a statement that can be true or false: "is the value of the Materials key available on the Resource aspect (R) from the upstream Function (id = 4) greater than or equal to 70?". If this statement is true, then the next value (or formula result) up to the colon (:) is returned, in this case the text "Execute Plan A". Otherwise, the remaining value or formula after the colon is returned, "Execute Plan B" (Figure 12).

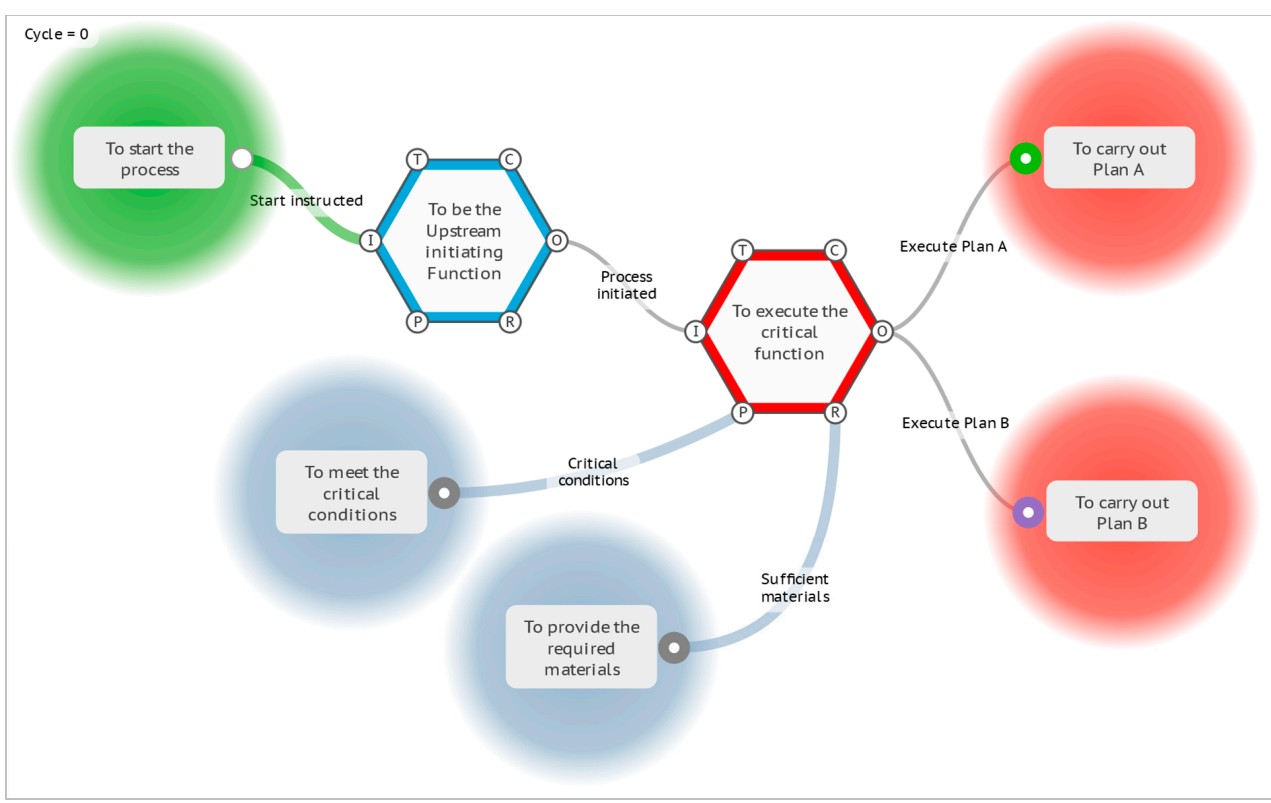

**Figure 9.** The default "Plan A or Plan B" model.

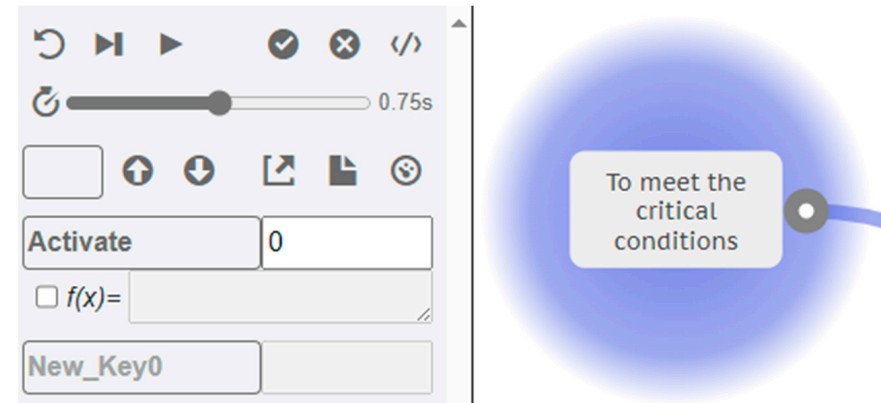

**Figure 10.** The Activate key with a value of 0.

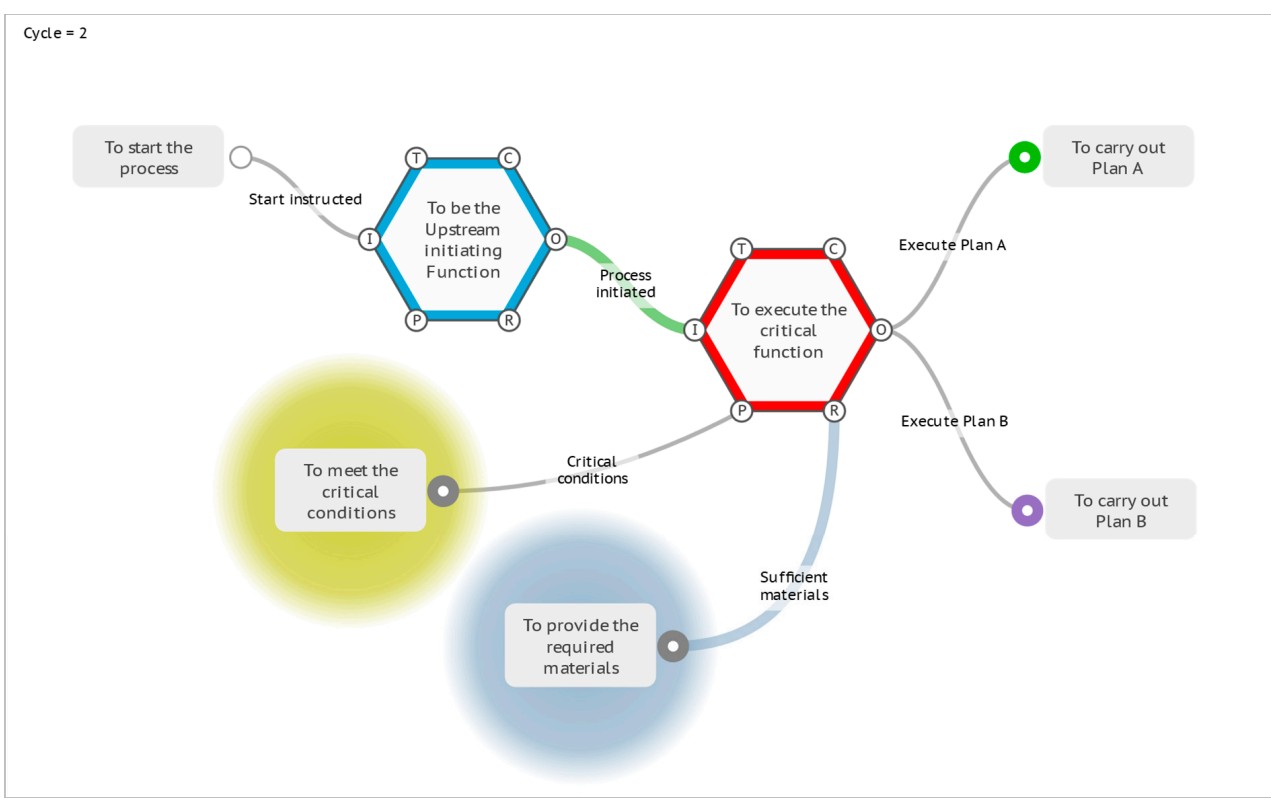

**Figure 11.** The "Plan A or Plan B" model with a precondition not activated.

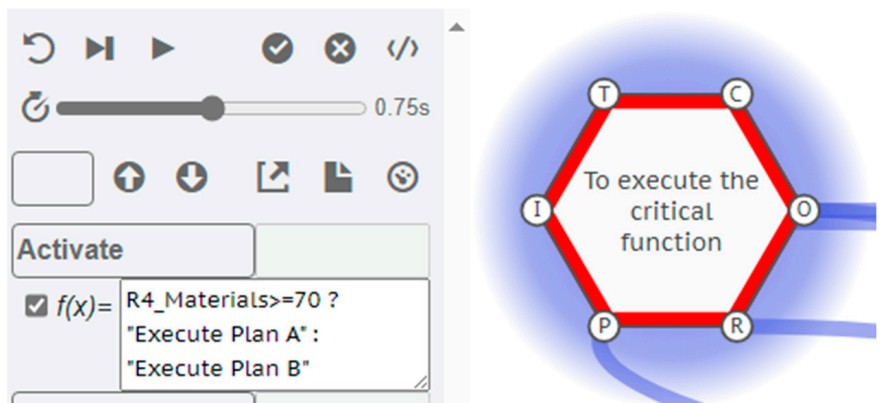

**Figure 12.** The Activate key with a conditional formula.

When the interpretation is run, only one Exit Function is activated on Cycle 3, <To carry out Plan A>, and the interpretation stops on Cycle 4 with no more Functions activated.

If the Materials key on the background Function <To provide the required materials> is changed to a value that is less than 70 and the interpretation re-run, then only the Exit Function <To carry out Plan B> will be activated.

### 4.2.3. Resilience Potentials

In creating a model, the FRAM method focuses primarily on describing the Functions, with the mutual couplings between Functions emerging from the increasing understanding of the Functions themselves. As a model becomes increasingly complex, it more often resembles a neural network rather than a linear process flow, and some Functions may be both upstream and downstream of others. Since Hollnagel published his Resilience Potentials in 2018 (to respond, to monitor, to learn, and to anticipate) [26], many users of

FRAM models have used these as a basis to describe the feedback and feedforward loops in their models. In some cases, like the Patient Management application described later on, they have been added to a model as new Functions, to intentionally create new feedback and feedforward loops and introduce resilience into an existing process.

A generic model of the resilience potentials is provided (Figure 13), as an example of how the FMI profile handles looping, and how the metadata can be used for iterative calculations.

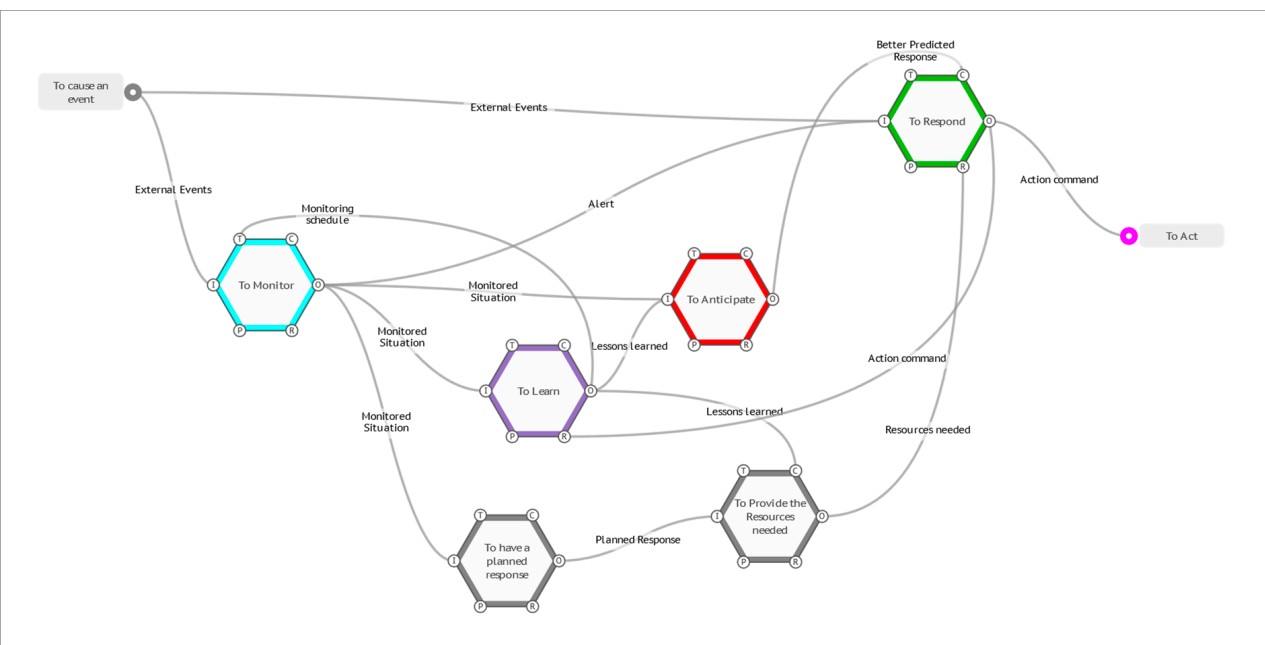

**Figure 13.** A generic FRAM model of the resilience potentials.

The model has an Entry Function <To cause an event>, that will activate on Cycle 0 of an FMI interpretation. There are two downstream Functions, <To Monitor> and <To Respond>, which would receive Inputs and potentially activate on Cycle 1. However, both Functions have other aspects that are coupled with other downstream functions, which in this case are also upstream of these two Functions. Because the default FMI profile requires that 'all' aspects are present before a Function produces an Output, the interpretation hangs at Cycle 1 and no further progress is visualised. This is a valid response, as one of the purposes of the FMI, as described previously, is to identify this type of mutual dependence, and to confirm if it is valid for the given model.

To continue the interpretation, it is necessary to consider the initial activation conditions for the Functions that are waiting for downstream aspects. Take the Function <To Respond>, for example, which has a Control aspect of [Better Predicted Response] and a Resource aspect of [Resources needed]. If it is initiated by the event on its Input, can it produce an Output if adequate controls and resources have not yet been established, or is there an initial unprepared reaction that will be later refined by downstream learning? An initial set of controls and resources can be added to the model as additional background Functions, and the FMI profile of the Function set to activate on 'any' Control aspect and 'any' Resource aspect, allowing it to activate if one or the other coupling on each of the aspects is present. For this example, we will take this as assumed, and more simply set the FMI profile on both the Control and Resource aspect to 'none', so that the Function can activate on the first instance with no other aspect couplings present so that it does not wait for all of its aspects. Similarly, we will also set the Time aspect on the Function <To Monitor> to 'none', representing an initial monitoring schedule that will be later refined by downstream learning.

When the FMI is run under these conditions, the interpretation progresses to Cycle 3 with the activation of the Exit Function <To Act>. If we were to advance the interpretation one more Cycle (4), the fourth resilience potential <To Anticipate> would activate for the first time and provide the Control aspect [Better Predicted Response] to the Function <To Respond>. Now, if the Entry Function activated again with a new [External Events] coupling, the model is prepared with more resources and controls for a better response. At this point, the FMI interpretation has shown that all Functions can activate and that all potential couplings are capable of being realised.

Next, we can model the second and subsequent recurring events by setting the FMI profile of the Entry Function to repeat every four Cycles, effectively pulsing the model at the same frequency that it takes to activate all Functions. Metadata can be used to represent and display the expected quality of the response, based on the number of times the model has iterated through the resilience potentials.

In a very simple case, we can show this by adding a Metadata key of OutputDisplay to each Function. For the Entry Function <To cause an event>, we set the starting value at 1. For all of the remaining Functions, we set them to the same formula, Aavg_OutputDisplay. This is a built-in formula that calculates the average of all of the OutputDisplay values on all of the aspects. If the aspect is not present, then the value will default to 0. For the <To Monitor> function on Cycle 1, the value coming from the upstream Entry Function is 1, while the Time aspect is not present and evaluates to zero, so the average that is then passed to the Output is 0.5. In this way, the calculated values start at 0.5 and approach 1 on subsequent cycles, as values from looping aspects are added to the calculations.

Finally, we set up the display results for the OutputDisplay key on an orange-to-blue scale from 0.5 to 1 (Figure 14).

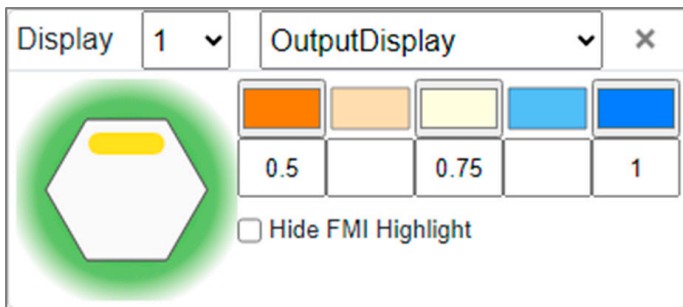

**Figure 14.** The FMV Display results setup for the OutputDisplay values.

The first time all Functions produce an output on Cycle 4, their results are all less than or equal to 0.5, and show orange. As the OutputDisplay values approach 1 on subsequent iterations of the model, the display results move from orange, through pale-yellow, to blue, with all Functions eventually displaying blue, such as the <To Respond> Function calculating an OutputDisplay value of about 0.9 on Cycle 16 (Figure 15).

In practice, this type of model could represent learning how to monitor for precursors, so that unwanted outcomes can be anticipated, and responses can be preventative or interventionary, as opposed to reactionary.

The principles of repeatedly cycling a model with loops can also be used for other types of iterative calculations, such as modelling steady-state conditions, optimising parameters, or even machine learning algorithms.

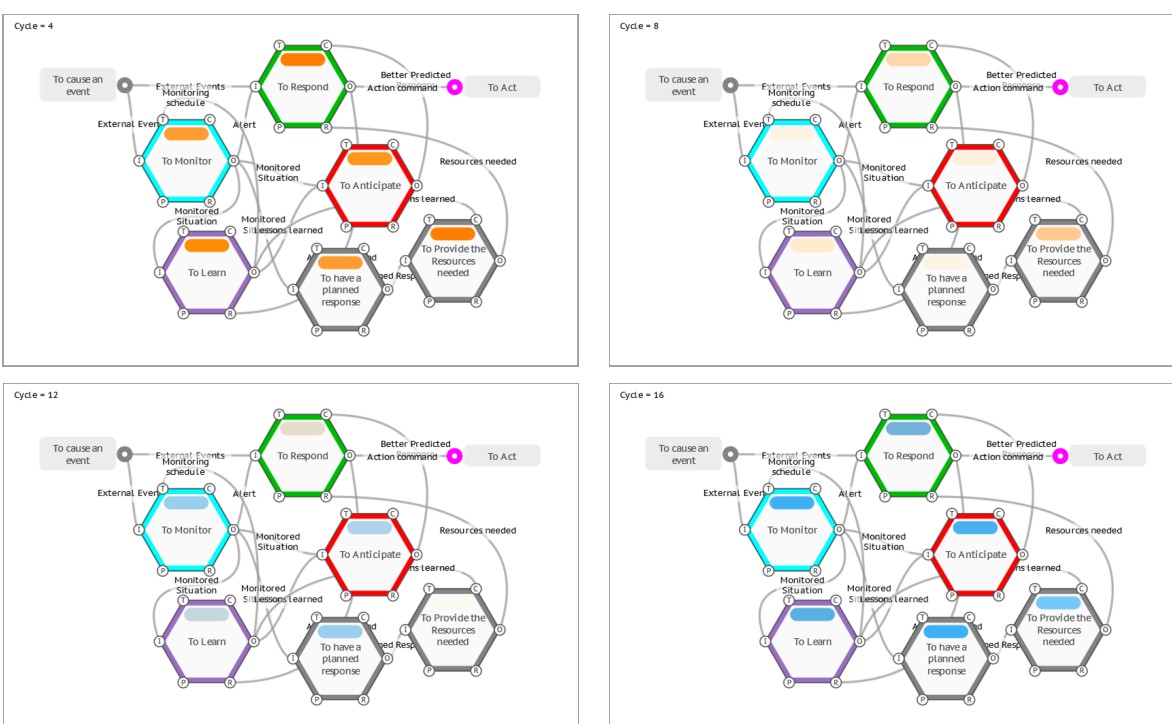

**Figure 15.** "Resilience Potentials" results on Cycle 4, 8, 12 and 16.

### 4.3. Current Applications of the Metadata

#### 4.3.1. Formula 1 Pit Stop

An interesting application of the Metadata option in FRAM was developed to explore its usefulness in assessing overall times taken for a process in which a series of independent, but linked, functions had to combine to complete the process. The example chosen was a to build a FRAM of a Formula 1 pit stop process and, specifically, the tyre changing instantiation. The results are detailed in the paper by Slater [27], and show an interesting confirmation of the applicability of the approach.

The process as prescribed (proscribed) by the official rules (Work as Imagined) was modelled, and predicted "average" times obtained of around 2.5 s. It was then noted that in actual racing scenarios (Work as Done), times as low as 1.8 s had been achieved. Careful examination of videos of actual situations were studied, and showed that the fastest teams were achieving the result by anticipating key events, such as moving the wheel guns towards the cars before they had stopped in their "boxes", so that the wheel nuts were being untightened before the car was jacked up; similarly, the teams were anticipating and releasing the cars as the wheel nuts were being tightened and before the jack had been fully lowered.

The validity of the analysis was confirmed by an actual racing incident in which this anticipation in releasing the car broke the leg of the front jack man, who did not have time to get out of the way. The rules were changed after that event, adding interlocks and monitoring, and the pit stop times then reverted to nearer the FRAM-predicted 2.5 s for the work, as "should have been done".

#### 4.3.2. Patient Management/Safety

Bed availability and assignment is a critical daily problem for hard-pressed hospital staff trying to cope with increasing demand against a background of reducing bed provision nationally. A morning meeting has to assemble all the staff responsible for operating and nursing the inevitably oversubscribed waiting lists, to allow them to update the meeting on the latest situation. The process involves the interaction and negotiation of numerous

considerations, to enable a consensus view of what is to be done. The next day exactly the same thing happens afresh.

A FRAM model of this process of bed assignment was built and analysed with medical professionals involved (MacKinnon [28]) It quickly became clear that although theoretically the meeting decided the (imagined) allocations for the day, there was a continuous process throughout the day of adapting to emerging situations, which gave them the resilience to cope not always "as imagined", or as decided in the meeting. The process is shown in Figure 16.

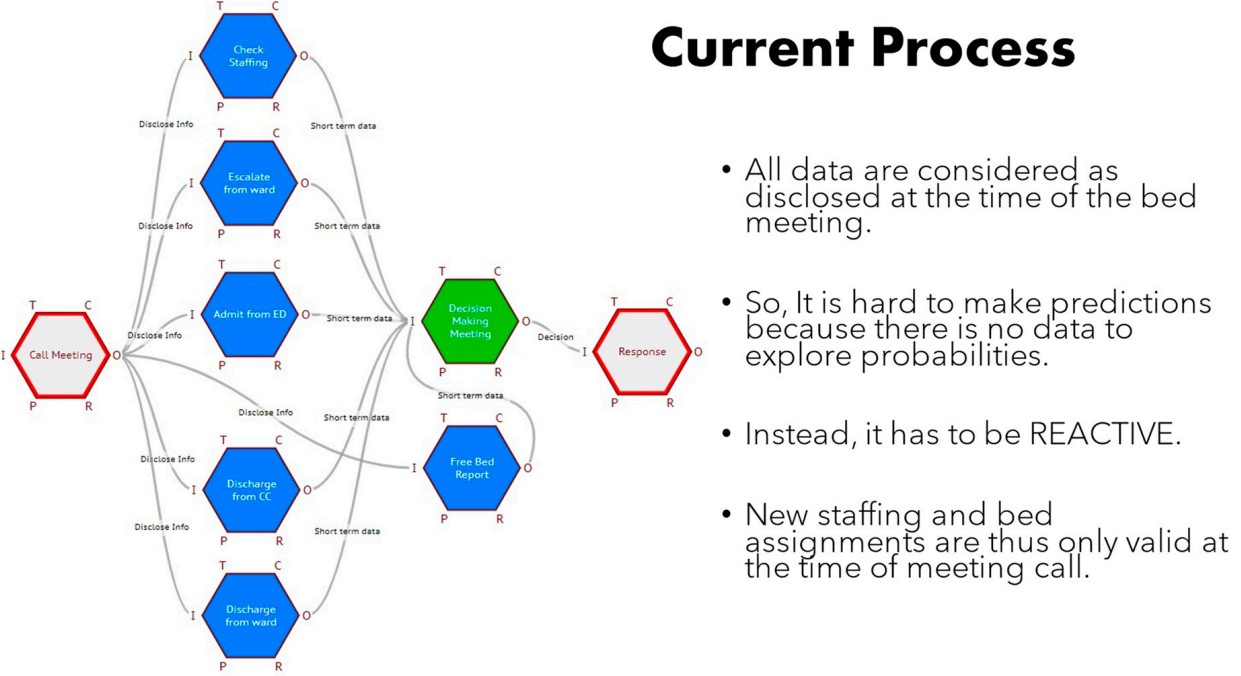

**Figure 16.** The current bed allocation system.

Everyone agreed that this was not only inefficient, meaning estimates always individually erred on the side of their own personal degrees of caution, but that it also resulted in elective-surgery patients (in this case children) being prepared for surgery, waiting around most of the day, only to be sent home to try another day. It was also apparent that there was no single source of integrated information, perhaps because it became out of date very quickly.

The FRAM analysis and capabilities allowed an alternative proposal (Figure 17).

All the data/information needed had already been recorded, and was recoverable from the hospital computer system as patient data records, and was not only retrievable, but available for predicting trends and learning patterns. The paper proposed using the FRAM model to visualise and track the bed data as metadata, through another option provided by Hill in the FMV, to import data from an external source. Thus, a new role (function) of Bed coordinator could now preprocess and continuously update all the relevant staff immediately with current and predicted demands and usages. It is further planned to utilise the FRAM with metadata to learn patterns of demand and availability, not just seasonally, but in response to learned precursors or "weak signals".

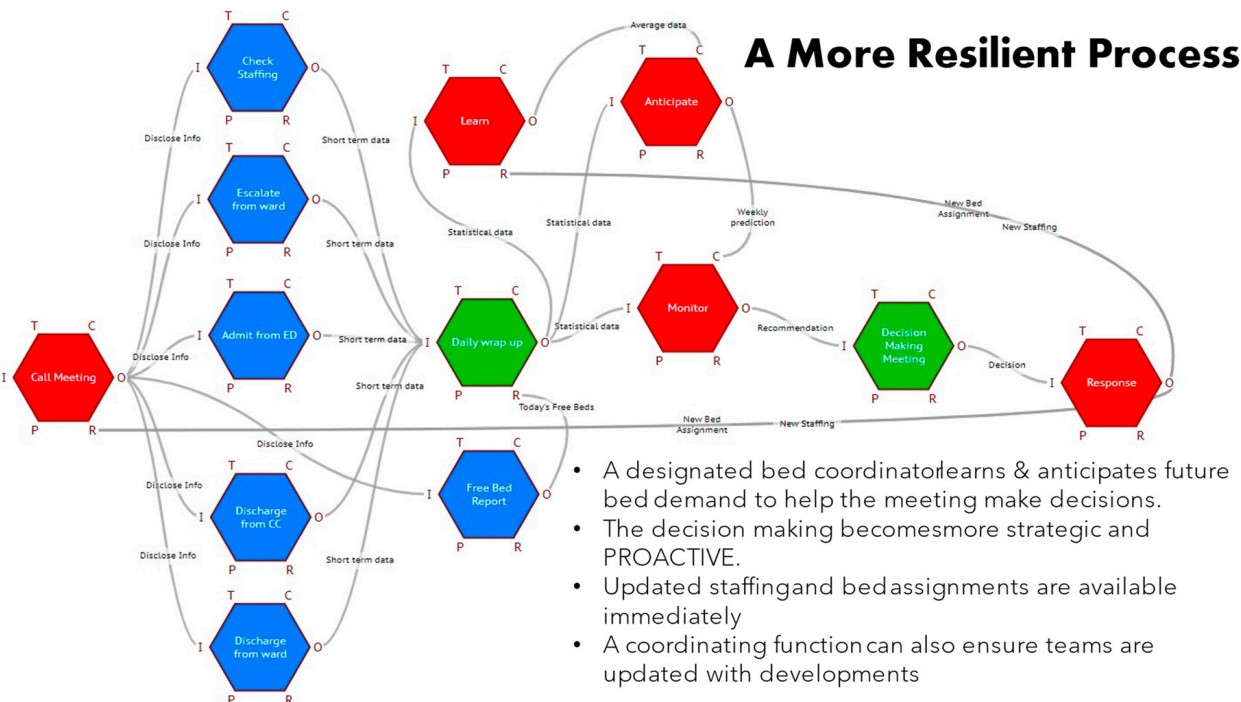

**Figure 17.** The alternative process suggested by the FRAM analysis.

## 5. Discussion

### 5.1. The Current Utility of the Approach

The introduction to this paper emphasised the "gap" in our armoury to analyse and predict the emergent behaviours recognised in the complex segment of the Cynefin framework. This emphasises that, in these systems, one cannot assume that modelling with rigid predetermined couplings applies, and there is a need to "probe" these systems to discover emergent behaviours. The main aim of this paper has been to establish firmly the academic bona fides and acceptability of the FRAM methodology, as more than just elaborate mind maps to be used in "safety differently" or SAFETY II arguments. The metadata development adds a quantitative dimension to its usefulness and one that has not been set out systematically and transparently before, to be peer reviewed and field tested!

The examples given here demonstrate the usefulness and versatility that this new metadata dimension for FRAM has made available to the system modeller. There has been criticism in the past that the FRAM lacked both the ability to be "validated" and to produce quantitative results from its application. While that might have been true for the early versions of the methodology, its continuous development, evolution and improvement from real applications and user feedback now clearly demonstrates that it can address these limitations.

One of the areas where the new utility of the FRAM modelling is receiving much attention is in using FRAM models to simulate system behaviours. For example, Mackinnon is looking at observed and video-recorded processes and simulations in CPR and the teamwork in critical care and resuscitation teams. There, they are using FRAM and metadata to make these observed simulations of medical treatments amenable to more quantitative analysis, by comparing and contrasting variations in operations among different teams and the effect of these variations on the effectiveness of the responses.

### 5.2. Further Possibilities Made Feasible

This ability to include machine learning opens up a number of possibilities in tracing more subtle interactions, particularly in systems in which human interactions dominate and in AI applications, with humans in the loop. These include the following:

1.  Human—Human systems (Iino [29]) There have been interesting papers paper on human–human interactions and FRAM modelling which allowed the pinpointing of the crucial success factors they provided in the cooperative operations in space manned missions, particularly those that were crucial factors in Apollo missions.
2.  Human robot systems (warehouses, robotic surgery (Adriaensen [30]) FRAM models of human–robot interactions have been studied in the interactions of robots and human workers in warehouse situations and the interaction of surgeons with robotic surgery systems.
3.  Self-driving cars , or auto autos (Grabbe [31]) Just the ability to process sensor data on the physical surroundings to predict and operate in real-world environments is not sufficient to design and program safe (fail-safe) behaviours for fully self-driving vehicles. These systems have to have the resilience to adapt to humans in the environments, who operate with a real-time, adaptive behaviour which, as we all know, is not necessarily commensurate with the highway code (often for legitimate circumstantial reasons). FRAM is being used to explore these interactions, in order to better incorporate them into control algorithms.
4.  Machine Learning (Nomoto [32]) Nomoto has used this metadata facility for FRAM to build a model of how the USD/CNY exchange rate depends on other Functions in the international economies. Using historical data, the model can learn and calibrate its metadata and equations to be able to predict forward movements of the currencies.
5.  Software system safety and security—AI? (Thurlby [33]) There is growing concern that the safety assurance of software systems is based on just assuring the individual components or subsystems (SAFETY I). Perhaps this is because they lack a methodology that can simulate the behaviour of a fully integrated system. This is becoming urgent as the pressures to include more and more AI controllers and autonomous agents are becoming increasingly irresistible.

### 5.3. The Future

The aim of the paper has been to develop and demonstrate a way that allows us to be able to model, analyse, simulate, and predict the behaviour of complex systems, and thus to assure ourselves that they are safe to operate. If we cannot model them adequately, then we will inevitably have difficulty managing them in practice. But having established its usefulness, a wide range of possible applications opens up. Some of these are suggested below.

#### 5.3.1. Process Simulation

Much of the work on complex sociotechnical systems relies on having a reliable and accurate "map" of the process being designed, its safety being assured, and it being commissioned, operated, changed, or decommissioned. This process map can then be systematically shared, reviewed, HAZOP'ed, etc., as a common visualisation of the systems and how it is intended and hoped to be employed. The ability to scrutinise, modify, and test different scenarios and to observe/predict accurately the consequences is then served by consulting this overall picture—e.g., a process flow diagram—and making observations and calculations based on the "picture" perceived. It is suggested here that the FRAM model offers not just an alternative static diagram, but that is a dynamic model, and as such is a more appropriate (simulated) basis for such systematic scrutiny.

#### 5.3.2. Digital Twins

Work is continuing on developing a capability to import live data from processes or operating systems as live feeds of Function "Values" into such a process model. This, together with the ability of the software to cycle the model continuously, could also allow the animation of the model. Using the FRAM model as a process simulation and this additional ability to make it "live" opens many opportunities to exploit it as a digital twin

of the process. This idea originates from the work of Nomoto [34] on the development of predictive FRAM models for business planning and management.

## 6. Conclusions

There is a common agreement on the need for, and an increasing demand for, the development of an acceptable/accepted method of modelling complex sociotechnical systems, in order to better understand and manage/operate them safely. The Functional Resonance Analysis Method (FRAM) has always had the ability to model complex systems in such a way that enables properties of systems as a whole to be explored. In particular, it is specifically designed to understand the combinations and impacts of real-world variabilities, which "cause" or enable unexpected resonances and emergent behaviours, to be uncovered and better understood. With this new ability to model quantitatively these behaviours and processes using metadata, FRAM has now evolved to a state where it is at least a candidate to be seriously considered [35].

This paper has demonstrated the basis of the approach using a software platform, which is freely available as an open-source code, so that it can be independently verified. It has further illustrated its usefulness in a number of practical applications. But the story does not end here; the methodology has opened the way to a series of possible uses, such as the safety and security of software systems, which with the advent of AI has made the need to keep a grip on the behaviour of these complex sociotechnical systems non-negotiable.

**Author Contributions:** Conceptualization, R.H. and D.S.; methodology, R.H.; software, R.H.; validation, R.H and D.S.; formal analysis, R.H.; investigation, D.S.; resources, R.H.; data curation, R.H.; writing—original draft preparation, D.S.; writing—review and editing, D.S.; visualization, R.H.; supervision, joint; project administration, D.S.; funding acquisition, No funding. All authors have read and agreed to the published version of the manuscript.

**Funding:** This research received no external funding.

**Data Availability Statement:** Data is currently restricted to the FRAM software Development group of academic researchers.

**Conflicts of Interest:** The authors declare that the research was conducted in the absence of any commercial or financial relationships that could be construed as a potential conflict of interest.

## Notes

1     https://en.wikiquote.org/wiki/William_Thomson (accessed on 7 March 2024).
2     INCOSE SE Vision 2020 (INCOSE-TP-2004-004-02, Sep 2007) OMG/).
3     The GitHub URL is https://github.com/functionalresonance (accessed on 7 March 2024).
4     Each stage of FMV software development was reviewed with Erik Hollnagel to ensure that the software always supported the original method and helped lead to its wide use and acceptance in teaching the FRAM method and visualising FRAM models.
5     https://www.york.ac.uk/assuring-autonomy/training/films/ (accessed on 7 March 2024).

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
