# Peer review of "Using a Metadata Approach to Extend the Functional Resonance Analysis Method to Model Quantitatively, Emergent Behaviours in Complex Systems"

_systems, doi:10.3390/systems12030090_

Round 1

Reviewer 1 Report (New Reviewer)

Comments and Suggestions for Authors

The research carried out in this paper extends the FRAM, which can realize the quantitative research on the safety of the system task process, and at the same time consider the system resilience to optimize the system task process, which makes a great contribution to the research on the safety of complex systems. If you can combine the following suggestions to improve the paper, you may make the paper better.

Suggestion 1

At present, there are many studies using formal models to describe the system and carry out safety analysis, such as modeling the system based on SysML, and conducting safety research on this basis. Therefore, it is suggested to add some differences between the current model-based system safety research methods and the methods described in this paper in Part 2, which can better explain the advantages of the methods proposed in this paper.

Suggestion 2

The research on resilience potential is the highlight of this paper,it is suggested to add some introduction on resilience modeling ideas in the first two parts.

Suggestion 3

At present, there are many studies on system resilience, and it may be necessary to add a part of explanation to explain the relationship between the content of this paper on resilience potential and other research about system resilience.

After study your paper, there are some questions about some parts that may need more instructions .

Question 1

This paper introduces a safety modeling method for complex systems, which can describe and predict the emergent behavior of the system during operation. However, whether the data requirement for this method is easily satisfied, because some parameters related to Functions are not always easy to obtain or need to be formulated manually.

Question 2

In the face of the task process of complex system, whether its complexity will cause the modeling complexity of the method described in this paper to be too high and difficult to carry out in application.

Question 3

The safety modeling method of complex system introduced in this paper seems to be aimed at the static task process, and the actual task may contain the dynamic change of the Function combination, so the explanation of this question may be insufficient.

Author Response

The comments and suggestions are valid and helpful. I have tried to answer them below.

Suggestion 1 - We have written quite a bit about the evolution and valid applicabilities of model-based system safety analysis methods. Perhaps if we included this summary (link below) as a reference, it would address the point?

https://www.researchgate.net/publication/377178359_Evolution_and_classification_of_safety_Models

Suggestion 2 – We are working with Erik Hollnagel on developing a more useable metric for resilience in systems. The current developments described, can enable us to model responding, Monitoring, Learning and Anticipating as separate functions or groups of functions which we can deal with directly. The suggestion anticipates the outcome of current research.

Suggestion 3 – see above.

You also raise valid questions, which we have tried to address below.

Question1 – The software makes it relatively easy to set and update the functional metadata. There are features included that makes it possible to access external data bases (e.g. Financial exchange rates and hospital bed data) and to import / monitor live feeds for digital twin applications.

Question 2 – Modelling complex systems by abstracting functional behaviours from component details is a great advantage in dealing with increasing complexity. Like Russian dolls, the level of abstractions (functions within functions) can be set to further offset seeing the wood for the trees.

Question 3 – As you know the FRAM is specially aimed at modelling dynamic systems, not fixed predetermined linear static node and edge and all FRAM models work by allowing the propagation to develop in a Markovian manner with the next step solely dependent on the current configuration.  It is a basic FRAM premise, so a reference to Hollnagel’s book should be sufficient?

Thankyou for your observations, I hope we have gone some way to satisfying the legitimate comments?

Reviewer 2 Report (New Reviewer)

Comments and Suggestions for Authors

The paper has the following serious problems: firstly, the whole paper is not refined enough. secondly, as a whole the paper does not revolve around the  topic question.  In addition, there are some minor errors in the text ,such as,  figure 8 is the table  actually.

Comments on the Quality of English Language

The writing should be refined and more academic.

Author Response

On reading and rereading these comments, my conclusions were that they must have been misfiled or incompletely uploaded as they seem incoherent, incomprehensible, and incomplete. So, I’m afraid that I struggled to grasp the points being made and feel unable to respond adequately?

Reviewer 3 Report (New Reviewer)

Comments and Suggestions for Authors

The article explains the FRAM methodology to create models, as well as the existing related software to visualize (FMV) and interpret (FMI) the created models. The authors try to explain through examples a standard way (the option Metadata) to insert equations or keys into the functions considered in FRAM models in order to obtain numerical results and to simulate such models quantitatively.

The article may be interesting for starting FRAM users, because it explains clearly the possibilities of the present state of the methodology and its future development. I am not a FRAM user. May be FRAM users could be better reviewers for this article.

Author Response

These comments were absolutely valid and admirably straightforward comments which I can only thank you for making them, and for taking the time to read our manuscript. I don’t think it needs a response?

Reviewer 4 Report (New Reviewer)

Comments and Suggestions for Authors

I am not a FRAM user, but I have done many systems simulations.  This paper appears to me to be almost a user's guide to FRAM.  It was hard for me to determine what is unique about FRAM.  How does it differ from methods that were used as much as 50 years ago, e.g. PERT.  It would help considerably to have a statement in the introduction saying what FRAM is and what it does, and why it is unique.  I would also like to know how it differs from some of the classic approaches to system simulation, e.g., Rasmussen's work, which is referenced, but goes back much further than 1997.

The section on "Current Modeling Approaches" covers several modeling approaches, but aggregates their comparison to FRAM only in the last paragraph, "What is needed and lacking in most of the current modeling approaches..."  I would like to better understand where FRAM fits in this set of modeling approaches. 

The Kelvin quote is paraphrased, but I'm not sure it is correct as stated.  There would seem to be a fundamental difference between modeling and measuring.  How do the authors define "model"?

"The general purpose of a model is to represent a selected set of characteristics..."  This may be one purpose, but perhaps not even the most common.  Do the authors agree with the quote?  Is this their purpose?

On page 4, "The purpose of FRAM..."  It would make the paper easier to read if a clear statement of the purpose of FRAM were provide on page 1.  And this statement is not very comprehensive, "to develop a description."  Also, on page 11, "the purpose of FRAM, which is a method to build a model,..."

The paper refers to a "gap" (Cynefin framework) identified in the Introduction, but I see no such reference.

Overall, the paper is very long, containing a lot of detail, but not providing a clear overview of its purpose and how it fits among other systems modeling tools.  What are FRAM's specific contributions?  I know that the authors will think that they do provide this, but it is embedded in detailed discussions that make it difficult to pick out.

There are no glaring errors that pop out to me.  But I wonder about definitions of some key words, such as model, safety, reliability, success and failure.  What definition of success and failure makes them equivalent?

Finally, I would ask, what is the purpose of this paper?  It's not a research paper.  It doesn't present research results, though it does present "validation" material.  Perhaps the authors could state the purpose of the paper and focus on that.  Is the purpose as stated in the abstract, "This paper describes how the FRAM methodology has now been extended..."

Author Response

Reflecting Reviewer 3’s comments, if a reviewer is unfamiliar with the more modern applications such as FRAM, I can understand the confusion. I also think you are looking at an earlier draft as I thought references to Dave Snowden’s Cynefin framework were no longer in the abstract?

I have written extensively about the evolution and applicability of modelling approaches and did not feel it necessary to regurgitate it. Perhaps if we include a reference to our latest summary (below) it will address these concerns for those who are unfamiliar with the field?

https://www.researchgate.net/publication/377178359_Evolution_and_classification_of_safety_Models

Round 2

Reviewer 2 Report (New Reviewer)

Comments and Suggestions for Authors

This paper described the FRAM methodology and its attributes. It also describes its implementation using an open-source software and how it really works in practice. 

Comments on the Quality of English Language

It is good at English writing.

Reviewer 4 Report (New Reviewer)

Comments and Suggestions for Authors

No additional comments.

This manuscript is a resubmission of an earlier submission. The following is a list of the peer review reports and author responses from that submission.

Round 1

Reviewer 1 Report

Comments and Suggestions for Authors

The work carried out by the authors presents an application and the interest of the FRAM method. The focus is on this framework. However:

- The title in no way reflects the work. The title is about the" Complex Systems Modeling: A systematic, Quantitative Systems Engineering Approach", Whereas the work dealt just at the beginning with the complex system briefly and then focused on the FRAM approach. The title needs to be revised and reworded.

- The abstract should be detailed and focused on the overall vision of the paper. References in the abstract should be avoided.

- For Complex Systems, there are also more recent works that deal with the complexity of systems and the systems approach, such as those by Garbolino and Gallab.

§  Gallab, M., Bouloiz, H., Tkiouat, M. 2019. Modeling and Simulation of Complex Industrial Systems”. LAMBERT Academic Publishing, ISBN: 978-620-0-31599-1.

§  Naciri, N., Tkiouat, M., 2015b. Complex system theory development. International Journal of Latest Researchin Science and Technology, Vol.4, Issue 6, pp. 93-103

§  Garbolino, E., Chery, J.P., Guanieri, F., 2009. Dynamic Systems Modelling to improve risk analysis in the context of seveso industries. Chemicals Engineering Transactions, vol. 17, pp. 373-378.

The introduction also needs to be revised, the problematic mentioned may be in a 2nd section, or in the 2nd part of the introduction, but first the general context needs to be mentioned and then the structuring of the paper.

- Sub-section 2.1 contains 2.1.1. only, the restructuring of the sub-sections needs to be reviewed.

-In section 3.2, you stated: "Perhaps two of the best are probably Patriarca 220 (2020), and Smith (2020)." The best compared to what? Have you defined any criteria for comparison?

The work is really an application of the FRAM method, with its areas of use, which is good, but unfortunately there's a missing link. The paper's objective is not very explicit.

Reviewer 2 Report

Comments and Suggestions for Authors

The manuscript as written reads like a tutorial on use of the FRAM methodology using the FMI tool; the scholarly contribution of the research described in this paper is not clearly evident. Specifically: (i) the specific shortcomings of other functional analysis methodologies (e.g. FFBDs, etc.) or process modeling methodologies which are addressed by FRAM are not addressed rigorously; it is not evident to me what was accomplished using this methodology which could not have been accomplished using SysML activity diagrams and parametric relations, for instance -- or any number of methods other than FRAM; (ii) at least one assertion is simply incorrect, such as the reference to Goldberg's SE toolbox document as a library of models, which it is not; rather, it is a handbook of techniques and methods; and (iii) the simple examples used to illustrate the FMI method fail to support the conclusions drawn that the tool can bring any unique capability to the fore in modeling complex systems. 

Reviewer 3 Report

Comments and Suggestions for Authors

General comments

- Multiple instances of language/phrasing that seems to have clear personal meaning to them, but which may not be as clear to the regular reader. Please see a few examples below. I suggest going through the entire manuscript to check and fix for this issue.

- Double quotation marks around certain terms where a clarification/explanation would have been more useful for the reader. Please see a few examples below. I suggest going through the entire manuscript to check and fix for this issue.

Lines 45-46

- Provide citations to your statements concerning increasing complexity and unpredictability.

Lines 46-47

The meaning of the sentence "“Artificial” enhancements seem actually to make them more opaque." is not clear. Please elaborate on it.

Line 56

- Provide the citation for Hollnagel here.

- Explain or provide a working definition for "complex system" and for "complicated system", both of which are terms you use in this work.

Line 64

- Whose limitations? Limitations pertaining to what? Clarify.

- What is "it" referring to? Clarify.

Line 70

- What is "it" referring to? Clarify.

Lines 73-177

- I would recommend looking into Object-Process Methodology (OPM) and its software implementation OPCloud, which together seem to address at least some of the challenges raised here with regard to modeling complex systems and to validating these models (including execution in OPCloud). I believe the paper would benefit from an explanation of how the proposed further refinement of FRAM would be superior to OPM in addressing the challenges with modeling complex systems. As an established methodology in systems engineering and an ISO standard, I do not think OPM can be overlooked when discussing this topic. 

Reference:

Dori, D. (2016). Model-based systems engineering with OPM and SysML (Vol. 15). New York: Springer.

One potential advantage of this approach over OPM (though I am not sure) is mentioned in your manuscript:

"In a conventional FRAM analysis, the possible effects of cooler water, or shorter times could be flagged as a variability to be noted. With the metadata we can now follow / predict the development and the quantitative effect of the ripples produced by this variability, right throughout the instantiation; and track any “emergence” of unexpected effects, or “resonances”."

Line 75

- Provide citations for your statement.

Line 97

- At the first instance of mentioning an acronym, write it out in full.

Line 132

- What do "ab" and "normal" mean in this context? Clarify.

Line 146

- Why is the word "barriers" placed in double quotation marks? Clarify. 

Figure 15

- I wonder, how would one avoid creating a "spaghetti model" with this methodology? Meaning, how will you manage the complexity of the model when modeling a very large/very complex system (nesting, decomposition, etc.)?

Figures 18 and 19

- If possible, please provide larger, higher resolution images for these figures. Some of the text is hard to read.

Line 803

- The use of exclamation mark is not required. It is not clear to me why it is needed here.

Comments on the Quality of English Language

Please consider having an external reader go through the manuscript for readability, flow, and clarity. I have provided a few specific examples below, but I suggest checking the entire manuscript and fixing these issues.

- Excessive use of commas in sentences, affecting reading fluency and comprehension. 

- Missing question marks at the end of sentences which clearly pose a question.

Lines 50, 53

- Missing a question mark. 

line 130

- Redundant parenthesis.

Lines 175, 216

- A superfluous comma before "the".

Line 743

- A superfluous comma before "was".

Line 745

- A superfluous comma before "that".